# Nonlinear Model Predictive Impedance Control of a Fully Actuated Hexarotor for Physical Interaction

**DOI:** 10.3390/s23115231

**Published:** 2023-05-31

**Authors:** Ran Jiao, Jianfeng Li, Yongfeng Rong, Taogang Hou

**Affiliations:** 1Beijing Key Laboratory of Advanced Manufacturing Technology, Faculty of Materials and Manufacturing, Beijing University of Technology, Beijing 100124, China; jiaoran@bjut.edu.cn (R.J.); lijianfeng@bjut.edu.cn (J.L.); 2School of Mechanical Engineering and Automation, Beihang University, Beijing 100191, China; ryf_2018@buaa.edu.cn; 3School of Electronic and Information Engineering, Beijing Jiaotong University, Beijing 100044, China

**Keywords:** impedance control, model predictive control, nonlinear control, physical interaction, unmanned aerial vehicle

## Abstract

In this paper, the problem of a fully actuated hexarotor performing a physical interaction with the environment through a rigidly attached tool is considered. A nonlinear model predictive impedance control (NMPIC) method is proposed to achieve the goal in which the controller is able to simultaneously handle the constraints and maintain the compliant behavior. The design of NMPIC is the combination of a nonlinear model predictive control and impedance control based on the dynamics of the system. A disturbance observer is exploited to estimate the external wrench and then provide compensation for the model which was employed in the controller. Moreover, a weight adaptive strategy is proposed to perform the online tuning of the weighting matrix of the cost function within the optimal problem of NMPIC to improve the performance and stability. The effectiveness and advantages of the proposed method are validated by several simulations in different scenarios compared with the general impedance controller. The results also indicate that the proposed method opens a novel way for interaction force regulation.

## 1. Introduction

Unmanned aerial vehicles (UAVs) have attracted increasing attention in recent years for their wide range of applications, including pesticide spraying [1], aerial photography [2], environmental monitoring [3], reconnaissance [4], and transportation [5]. In addition to these non-contact tasks, there are increasing demands on contact-based operations such as industrial contact inspection [6], window-cleaning [7], peg-in-hole assembly [8], and aerial grasping [9]. Physical interactions between the UAV and environments is usually required in contact-based tasks, which is quite challenging for the controller design since more constraints should be considered and the environments may be undesirable. Additionally, the complexity of aerodynamics and the intrinsic instability further increase the difficulties.

To perform physical interaction with the environment, there have been several studies on the classical under-actuated UAV (UUAV) equipped with either a fixed tool [10] or a degree-of-freedom (DoF) robot arm [11,12]. They have reached remarkable performances when dealing with simple contact tasks. However, they lack dexterity due to the underactuation of the platform. Additionally, the coupled rotational and translational dynamics result in the limited lateral force magnitudes and disturbance rejection capabilities. Moreover, they may not be able to maintain stability when unexpected situation occurs (e.g., when the contact force suddenly disappears). For the aforementioned reasons, the fully actuated UAVs (FUAVs) for aerial interactions have emerged to address the problems in recent years. Compared with the UUAVs, the rotational and translational dynamics of FUAVs can be decoupled, which brings convenience to the controller design. A broad class of mechanical structures of FUAVs were reviewed in [13]. The discussion of the mechanical structures is beyond the scope of this paper, so the control strategies will be focused on the following.

To achieve the aerial physical interaction with the environments, and in contrast to the traditional proportional–integral–derivative (PID) controller [14] used for UAV attitude stability, the fractional order PID (FOPID) [15,16] built for dealing with actuator faults, wind effects, and some other disturbances, the controllers have to simultaneously deal with both force and motion control. In the work by [17], a hybrid controller is proposed with force control in 1 DoF for interactions and motion control for the other 5 DoFs, which is a straightforward and effective means of physically interacting. However, it is not compatible with unpredictable interactions. Therefore, a more suitable strategy is to design the controllers as inherently compatible with the environmental interactions. Among the existing interactive control methods, impedance and admittance control [18,19,20] are the most popular and widely used in various fields, particularly in robots interacting with humans/environments, which are more suitable for physical interactive control than the PID class method that is not model-based. The impedance method is exploited in [21] for a tethered UAV exerting physical human–robot interaction. Lovro et al. in [22] proposed using an impedance controller coupled with an online stiffness estimation algorithm for UAV to exert aerial inspection, applying regulated force via the end-effector to the contact surface. However, these methods are particularly designed for UUAVs, which are difficult to generalize to FUAVs. For FUAVs, a flying end-effector paradigm by means of the admittance filter is proposed in [23] to achieve 6D interaction, and the paradigm is able to control the full 6D pose and exert a full-wrench with a rigidly attached end-effector. Moreover, an impedance control (IC) scheme is proposed in [24] for aerial robotic manipulators based on a multi-layer architecture. To allow compliance in certain degrees of freedom while maintaining accurate trajectory tracking and disturbance rejection abilities in others, an axis-selective impedance controller was proposed in [25] and extended to a variable axis-selective impedance controller in [26]. The aforementioned methods may all obtain satisfactory performances under normal situations; however, they lack consideration of the constraints in their controller design. In real robotic systems, constraints always exist, such as actuator saturation, speed limits, force limits, etc. These limits are the key parts for guaranteeing safety and stability, especially in the physical interaction process.

To deal with constraints, model predictive control (MPC), as a model-based method that works for both linear and nonlinear systems, and even hybrid systems, is widely employed in many applications. The MPC can simultaneously optimize towards the optimal tracking of the reference trajectory, satisfy the input and state constraints, and maintain robust performance during a finite horizon. Manzoor et al. in [27] proposed a robust MPC framework for the trajectory tracking of a ducted fan aircraft subject to different uncertainties and external disturbances. A novel nonlinear information fusion model predictive control (NIFMPC) approach is proposed in [28] for a UUAV under external wind disturbance. The nonlinear model predictive control (NMPC) methods were extended to FUAVs or overactuated UAVs recently. In the work of [29], an NMPC method is applied for an overactuated UAV to solve trajectory tracking problems under constraints. A NMPC method is applied to multi-rotor aerial systems in [30] with arbitrarily positioned and oriented rotors considering the actuator constraints. However, the aforementioned MPC is not designed for interaction control. There have been some other research works on the MPC for physical interactions recently. In [31,32], the NMPC is combined with nonlinear moving horizon estimation (NMHE) for an aerial physical interaction task. However, the compliant behavior is obtained by the mechanical structure of the attached tool. Thus, the compliant characteristics are not feasible to adjust compared with software methods. In the work of [33], a model predictive impedance controller (MPIC) is presented to allow compliant behavior while respecting practical robotic constraints for physical interaction between the robot arm and environment. However, this method is only applied to a base-fixed robot arm system. In the flying system with a floating base, the situation is more complicated. Moreover, Nava et al. in [34] presented an optimization-based method to control aerial manipulators in physical contact with the environment considering hard constraints, although this idea is very similar to the one in this paper, the desired dynamics are provided by a PID controller which cannot obtain compliant behavior.

One critical problem for MPC is the requirement for an accurate dynamic model. There are various methods to address this problem now, e.g., HMHE [31], extended Kalman filter (EKF) [29], and disturbance observer (DO) [35]. In fact, these methods can also be employed for the measurement of an external wrench (force and torque) for interaction control without force sensors. These estimators are also called wrench estimators in some literature. The wrench estimators used for interaction control in [23,25] have the same structures as DOs; however, in general, the estimated wrench provided by DO is the sum of the physical interaction wrench, the aerodynamic wrench, a fault wrench, and other disturbances [36]. The lumped estimated wrench can be used in anti-disturbance control such as [37]. In the field of the physical interaction control, separating the physical interaction wrench from other disturbances is another popular topic which is beyond the scope of this paper. Thus, in this work, the DO will be employed for both wrench estimation and dynamic model compensation for MPC under the assumption that the physical interaction wrench covers the majority of the lumped estimated wrench. Similar assumptions can be found in [38] where the generalized momenta-based DO and the EKF algorithm are exploited in the joint torque estimation.

Another critical problem for MPC is the weight adaptation problem. In the traditional MPC, the weights of the cost function are usually fixed during the control process after tuning by human experts; however, fixed weights may not work well in some complex applications. To overcome this problem, Taherian et al. in [39] derived the transformation of a linear quadratic regulator weighting matrix using a discrete algebraic Riccati equation, and then an MPC controller adopting the idea of a discrete time linear quadratic tracking problem was proposed. Moreover, an exponential weight is introduced in [40] to solve the mathematical problem inherent in the MPC path tracking controller. In the work of [41], a weight adaptive strategy based on the fuzzy logic algorithm is proposed in the NMPC cost function to adaptively prioritize vehicle dynamics or energy efficiency, depending on the driving conditions. In the physical interaction process, the FUAVs will also go through different states or situations that require different weights. Thus, designing an adaptive weight strategy is also a challenging problem.

Based on the above observations and inspired by [33,34], a novel control scheme using nonlinear model predictive impedance control (NMPIC) on the basis of a weight-adaptive strategy together with DO for physical interaction with an FUAV considering constraints is proposed in this paper. The main contributions are listed as follows.

A novel control scheme combining NMPC and IC is proposed to handle the constraints and maintain the compliant behavior for physical interactions with a FUAV.A weight-adaptive strategy is proposed for NMPIC to handle the inherent conflict between the reference signals and the constraints.Comparative simulations are conducted to verify the effectiveness and advantages of the proposed method in different scenarios.The proposed method also opens a new way for interaction force regulation.

This paper is organized as follows. The system model together with IC and DO designed based on the model are introduced in Section 2. Dependent on the IC and DO, the proposed NMPIC method with a weight adaptive strategy for the cost function is proposed in Section 3. Comparative simulation results and discussions are given in Section 4. Finally, conclusions and perspectives are presented in Section 5.

## 2. Problem Formulation

To describe the system, the model of the platform is firstly established. After that, a model-based impedance controller and disturbance observer are introduced. The controller and observer will be exploited as the reference dynamics of the NMPC methods presented in the next section.

### 2.1. System Model

To describe the dynamics of the platform, two coordinate frames are introduced, the body frame $\left(O_{B},X_{B},Y_{B},Z_{B}\right)$ and the world frame $\left(O_{W},X_{W},Y_{W},Z_{W}\right)$. Figure 1 illustrates the FUAV platform used in this work. The model is transformed from a classical hexarotor with six equally spaced arms around the $Z_{B}$ axis. Each propeller is tilted by a fixed angle around the corresponding arm. A manipulation tool is rigidly attached to the platform body with a length *L* from the tip to the center of mass. The attached tool is assumed to pass through the center of mass of the robot such that there is no offset. Additional mass will be also added for counter balance with the attached tool. Using the Newton–Euler formalism, the simplified dynamic model with respect to the center of mass of FUAV can be expressed as follows.
(1)$$\begin{array}{c} m\ddot{r}=-mge_{3}+RF+RF_{e}\\ \dot{\Psi }=W\omega \\ J\dot{\omega }=S\left(J\omega \right)\omega +\tau +\tau_{e} \end{array}$$
where $r={\left[x,y,z\right]}^{T}$ denotes the position of the platform with respect to the world frame. $\Psi ={\left[\varphi ,\theta ,\psi \right]}^{T}$ and $\omega ={\left[\omega_{x},\omega_{y},\omega_{z}\right]}^{T}$ represent the attitude and the angular velocity of the platform expressed in the body frame, respectively. $S\left(\cdot \right)$ denotes the skew-symmetric matrix operator. *m* and $J=diag\left(J_{xx},J_{yy},J_{zz}\right)$ are the total mass and moment of inertia of the platform, respectively. *g* is the gravity constant. $F_{e},\tau_{e}\in R^{3}$ denotes the external force and torque expressed in body frame, respectively. $F,\tau \in R^{3}$ are the total thrust force and torque generated by the six tilted propellers, respectively. $R\in \mathrm{SO}\left(3\right)$ is the rotation matrix expressed with Euler angles Ψ. W is defined as
(2)$$W=\left[\begin{array}{ccc} 1 & \sin\phi \tan\theta & \cos\phi \tan\theta \\ 0 & \cos\phi & -\sin\phi \\ 0 & \sin\phi \sec\theta & \cos\phi \sec\theta \end{array}\right]$$

The usage of Euler angles will certainly cause singularity when $\theta =\pm \frac{\pi }{2}$; however, in our case, the angles will be limited by the constraints to avoid the singularity, which can be seen in the next section. Thus, it is sufficient to exploit Euler angles here. For compact expression, system (Equation 1) can be rewritten in the following form.
(3)$$M\dot{\upsilon }+C\upsilon +G=\tilde{u}+\tilde{d}$$
with
$$M=\left[\begin{array}{cc} mI_{3} & 0\\ 0 & J \end{array}\right],C=\left[\begin{array}{cc} 0 & 0\\ 0 & -S\left(J\omega \right) \end{array}\right]$$
$$G=\left[\begin{array}{c} mge_{3}\\ 0 \end{array}\right],H=\left[\begin{array}{cc} R & 0\\ 0 & I_{3} \end{array}\right]$$
$$\tilde{u}=Hu,\tilde{d}=Hd$$
where $\upsilon ={\left[{\dot{r}}^{T},\omega^{T}\right]}^{T}$ is the generalized velocities. It is worth noting that $\dot{r}$ is expressed in the world frame and ω is expressed in the body frame. $u={\left[F^{T},\tau^{T}\right]}^{T}$ is the control wrench and $d={\left[F_{e}^{T},\tau_{e}^{T}\right]}^{T}$ is the external wrench. $I_{3}\in R^{3\times 3}$ is the identity matrix and $e_{3}={\left[0,0,1\right]}^{T}$ is the unit vector around the z axis.

To properly allocate the control wrench to each propeller, the allocation matrix is briefly introduced here.
(4)$$u=\left[\begin{array}{c} \sum_{i=1}^{6}f_{i}\\ \sum_{i=1}^{6}\left(p_{i}\times f_{i}+\tau_{i}\right) \end{array}\right]=AT$$
with
$$\begin{array}{c} f_{i}=c_{f}T_{i}R^{{}^{\prime }}e_{3},\\ \tau_{i}={\left(-1\right)}^{i-1}c_{\tau }T_{i}R^{{}^{\prime }}e_{3},\\ R^{{}^{\prime }}=R_{z}\left(\left(i-1\right)\frac{\pi }{3}\right)R_{x}\left({\left(-1\right)}^{i-1}\alpha \right),\\ p_{i}=R_{z}\left(\left(i-1\right)\frac{\pi }{3}\right)l+R_{x}\left({\left(-1\right)}^{i-1}\alpha \right)d,\\ i=1,\ldots ,6 \end{array}$$
where $f_{i},p_{i},i=1,2,\ldots ,6,$ are the force generated by the *i*th propeller and the position of each propeller expressed in body frame, respectively. $c_{f}$ and $c_{\tau }$ are the motor constant and the moment constant, respectively. α is the tilted angle around the the corresponding arm where the rotor is installed. *l* and *d* are the arm length and the position offset from the arm, respectively. $T={\left[T_{1},\ldots ,T_{6}\right]}^{T}$ denotes the thrust generated by each propeller and A is the resulting allocation matrix. More details can be found in [23].

### 2.2. Impedance Control

To design the impedance controller, the close-loop dynamics should be designed to perform compliant behavior, formulated as
(5)$$M_{\upsilon }\dot{\upsilon }+D_{\upsilon }e_{\upsilon }+K_{\upsilon }e_{p}=\tilde{d}$$
with
$$\begin{array}{c} e_{r}=r-r_{d},e_{\dot{r}}=\dot{r}-{\dot{r}}_{d}\\ e_{R}=\frac{1}{2}{\left(R_{d}^{T}R-R^{T}R_{d}\right)}^{\vee },e_{\omega }=\omega -R^{T}R_{d}\omega_{d}\\ e_{p}={\left[e_{r}^{T},e_{R}^{T}\right]}^{T},e_{\upsilon }={\left[e_{\dot{r}}^{T},e_{\omega }^{T}\right]}^{T} \end{array}$$
where $e_{r}$, $e_{R}$, $e_{\dot{r}}$, and $e_{\omega }$ are the position, rotation, velocity, and angular velocity tracking errors, respectively. $r_{d}$, ${\dot{r}}_{d}$, $R_{d}$, and $\omega_{d}$ are the desired position, velocity, rotation, and angular velocity, respectively. ${\left(\cdot \right)}^{\vee }$ denotes the unskew operation. $M_{\upsilon }$, $D_{\upsilon }$ and $K_{\upsilon }\in R^{6\times 6}$ are the positive definite diagonal matrices representing the desired apparent inertia matrix, damping matrix and stiffness matrix, respectively. By combining (Equation 3) and (Equation 5), the impedance controller can be derived as
(6)$${\tilde{u}}^{*}=\left(MM_{\upsilon }^{-1}-I_{6}\right)\hat{d}-MM_{\upsilon }^{-1}\left(D_{\upsilon }e_{\upsilon }+K_{\upsilon }e_{p}\right)+C\upsilon +G$$
where $\hat{d}$ is the external wrench estimated by the disturbance observer, which will be introduced in the following. The relationship between ${\tilde{u}}^{*}$ and $u^{*}$ is ${\tilde{u}}^{*}=Hu^{*}$.

### 2.3. Disturbance Observer

To estimate the external wrench, a widely used momentum-based disturbance observer is introduced here. For a unified form, the disturbance observer can be constructed as
(7)$$\left[\begin{array}{c} M\dot{\hat{\upsilon }}\\ \dot{\hat{d}} \end{array}\right]=\left[\begin{array}{c} -C\upsilon -G\\ 0 \end{array}\right]+\left[\begin{array}{c} \tilde{u}\\ 0 \end{array}\right]+L\left(M\upsilon -M\hat{\upsilon }\right)$$
where $L={\left[L_{1},L_{2}\right]}^{T}\in R^{12\times 6}$ with $L_{1}=2diag\left(\omega_{o1},\ldots ,\omega_{o6}\right)$ and $L_{2}=diag\left(\omega_{o1}^{2},\ldots ,\omega_{o6}^{2}\right)$. $\omega_{i},i=1,2,\ldots ,6$ denotes as the observer bandwidth. $\hat{\upsilon }$ and $\hat{d}$ are the estimation of the generalized velocity υ and the external wrench $\tilde{d}$, respectively. Similar structures for the exploited disturbance observer can be found in [26].

## 3. Nonlinear Model Predictive Impedance Control

### 3.1. NMPIC Design

To design a control strategy that can simultaneously handle the constraints and maintain the compliant behavior, the NMPC method is exploited to be combined with the impedance controller. When the FUAV is interacting with the environment within the constraints, the controller should have the same performance as the impedance controller. However, when one of the constraints is reached or some unexpected circumstances happen, the controller should be able to maintain stability within the constraints. In order to achieve this goal, the reference control input at time $t_{k}$ is set to be ${\tilde{u}}^{*}\left(t_{k}\right)$. This is also denoted as the virtual input of an outer loop in [34]. The real control input at time $t_{k}$ of the impedance controller is set to be
(8)$$\tilde{u}\left(t_{k}\right)={\tilde{u}}^{*}\left(t_{k}\right)$$

From another point of view, the control law (Equation 8) is equal to an NMPC problem with the following quadratic cost function without constraints.
(9)$$\min_{\tilde{U}}\sum_{i=k}^{k+N-1}{\left(\tilde{u}\left(t_{i}\right)-{\tilde{u}}^{*}\left(t_{i}\right)\right)}^{T}\Gamma \left(\tilde{u}\left(t_{i}\right)-{\tilde{u}}^{*}\left(t_{i}\right)\right)$$
where $\Gamma =diag\left(\gamma_{1},\ldots ,\gamma_{6}\right)\in R^{6\times 6}$ is a positive diagonal weighting matrix and $\tilde{U}={\left[{\tilde{u}}^{T}\left(t_{k}\right),\ldots ,{\tilde{u}}^{T}\left(t_{k+N-1}\right)\right]}^{T}$. The optimal problem (Equation 9) is solved over a finite horizon $\Delta t=NT_{s}$ of *N* steps with $T_{s}$ being the time step. It is obvious that the solution of the (Equation 9) without any constraints is exactly (Equation 8) when only the optimal control at current time $t_{k}$ is applied. However, in real physical interactions, some constraints exist objectively based on the physical laws, such as the actuator constraints and the model constraints. Other constraints for ensuring the safety in case that some unexpected situations occur during the process of physical interaction should also be considered, such as position constraints and velocity constraints. In this paper, the following constraints are considered.

Firstly, the model constraints should satisfy (Equation 1) with the state variables selected as $X={\left[r^{T},{\dot{r}}^{T},\Psi^{T},\omega^{T}\right]}^{T}$. In a real system, the model parameters cannot be accurately measured and some unmodeled dynamics may occur due to the simplicity in the modeling process. Therefore, the external wrench cannot be accurately predicted based on model. In this case, the external wrench is replaced by the estimated wrench provided by the DO. However, even with DO, only the current external wrench can be known, while the external wrench in the future remains unknown. To deal with this, we assume that the external wrench changes slowly during the prediction horizon. Thus, the estimated wrenches will be regarded as constants during the prediction period. To summarize, the model constraints are formulated as follows.
(10)$$\left\{\begin{array}{c} \dot{X}\left(t_{m+1}\right)=f\left(X\left(t_{m}\right),\tilde{u}\left(t_{m}\right),\hat{d}\left(t_{k}\right)\right)\\ X\left(t_{k}\right)=\mathrm{current}\;\mathrm{state} \end{array}\right.$$
where m=k,…,k+N−1 and $f\left(\cdot \right)$ is the discrete form of system (Equation 1).

Next, since physical actuators are bound to have their limits in the rotational speeds, the generated forces should also have boundaries. One can construct the actuator constraints as
(11)$$T_{\min}\leq T\leq T_{\max}$$
where $T_{\min}$, $T_{\max}$ are the minimum and maximum thrusts generated by each propeller. However, this is not intuitive during the physical interaction process, where the composite generated wrench should be limited. Thus, instead of using inequality (Equation 11), the following wrench constraints are used.
(12)$$u_{\min}\leq u\leq u_{\max}$$
where $u_{\min}$ and $u_{\max}$ are the minimum and maximum wrenches which have an indefinite sign.

Moreover, to show the power of the NMPC method, the constraints of the state variables X are also added.
(13)$$X_{\min}\leq X\leq X_{\max}$$

These constraints indicate that the positions, velocities, angles, and angular velocities are all limited. In this case, the angles will be limited to avoid meeting the singularity.

To conclude, the proposed NMPIC method is formulated as
(14)$$\begin{array}{c} \min_{\tilde{U}}\sum_{i=k}^{k+N}{\left(\tilde{u}\left(t_{i}\right)-{\tilde{u}}^{*}\left(t_{i}\right)\right)}^{T}\Gamma \left(\tilde{u}\left(t_{i}\right)-{\tilde{u}}^{*}\left(t_{i}\right)\right)\\ st.\left\{\begin{array}{c} \dot{X}\left(t_{m+1}\right)=f\left(X\left(t_{m}\right),\tilde{u}\left(t_{m}\right),\hat{d}\left(t_{k}\right)\right)\\ X\left(t_{k}\right)=\mathrm{current}\;\mathrm{state} \end{array}\right.,\;\;\;\begin{array}{cc} m=k,\ldots ,k+N-1 & \end{array}\\ \;\;\;\;\;u_{\min}\leq u\left(t_{m}\right)\leq u_{\max},\;\;\;\;\;\;\;\;\;m=k,\ldots ,k+N\\ \;\;\;\;\;X_{\min}\leq X\left(t_{m}\right)\leq X_{\max},\;\;\;\;\;\;\;\;\;m=k,\ldots ,k+N \end{array}$$

### 3.2. Weight Adaptive Strategy

In the traditional NMPC method, the parameters of the weighting matrix Γ are tuned based on human expert knowledge, which is time-consuming and cumbersome. Moreover, the weighting matrix is usually set to remain fixed all through the control process. However, in our case, this does not work for the reason that there is an inherent conflict between the reference signals and the constraints. The control force in the x axis will be taken as an example for explanation. Consider that a constraint limits the force in x axis in a certain range. Since the reference control input provided by (Equation 6) has no limit and the cost function is designed to make the real control input as close to the reference control input as possible. As a consequence, a large cost will be generated in the x axis to prevent the constraint from being conceded. In contrast, the costs in other five channels become relatively less, resulting in not following the reference control inputs of these channels. This may cause large tracking errors in other channels and even stability problems. To address this problem, a weight adaptive strategy is proposed in this work.

In the weight adaptive strategy, the weighting matrix Γ is the composition of two parts: the base weighting matrix $\Gamma_{b}$ and constraints-related weighting matrix $\Gamma_{c}$, formulated as,
(15)$$\Gamma =\Gamma_{b}\Gamma_{c}$$

The base weighting matrix $\Gamma_{b}=diag\left(\gamma_{b1},\ldots ,\gamma_{b6}\right),\gamma_{bi}>0,i=1,\ldots ,6$ is the same as the fixed weighting matrix in the traditional NMPC. The constraints-related weighting matrix is designed to satisfy $\Gamma_{c}=diag\left(\gamma_{c1},\ldots ,\gamma_{c6}\right),\gamma_{ci}\in \left(0,1\right],i=1,\ldots ,6$. The design of $\Gamma_{c}$ should meet the following requirements. When the reference controls can be reached within the constraints, there is no need to use $\Gamma_{c}$ so that all the parameters in $\Gamma_{c}$ should be set to 1. When the reference controls cannot be reached within the constraints, the parameter(s) associated with the corresponding channel(s) in $\Gamma_{c}$ should decrease according to some strategies in order to balance the other costs. A simple strategy is given as
(16)$$\begin{array}{c} \gamma_{ci}=\prod_{j=1}^{n}w_{ji},\;\;\;\;\;\;i=1,\ldots ,6\\ w_{ji}=\left\{\begin{array}{cc} \varepsilon_{j} & q_{j}\leq \alpha_{j}\\ \frac{1-\varepsilon_{j}}{\Delta_{j}}\left(q_{j}-\alpha_{j}\right)+\varepsilon_{j} & \alpha_{j}<q_{j}<\alpha_{j}+\Delta_{j}\\ 1 & \alpha_{j}+\Delta_{j}\leq q_{j}\leq \beta_{j}-\Delta_{j}\\ -\frac{1-\varepsilon_{j}}{\Delta_{j}}\left(q_{j}-\beta_{j}\right)+\varepsilon_{j} & \beta_{j}-\Delta_{j}<q_{j}<\beta_{j}\\ \varepsilon_{j} & q_{j}\geq \beta_{j} \end{array}\right. \end{array}$$
where $\alpha_{j}\leq q_{j}\leq \beta_{j}$ with $q_{j}\in X_{d}\cup u^{*}$ is one of the inequality constraints in (Equation 12) and (Equation 13). The subscript j denotes the *j*th constraint that relates to the corresponding control channel. *n* is the total number of constraints that correspond to the *i*th channel. The curve of $w_{ji}$ with respect to $q_{j}$ is illustrated in Figure 2.

For each channel, the corresponding relationship is given in Table 1.

To further explain the corresponding relationship, we will take the force control regarding the x axis as an example. In this case, $q_{j}$ is specified as $q_{1}=x_{d}$, $q_{2}={\dot{x}}_{d}$, and $q_{3}=u_{1}^{*}$. Then, the corresponding $w_{j1}$ and $\gamma_{c1}$ are computed by expressions given in (Equation 16). It can be seen that only the position, velocity, and force constraints regarding the x axis are assumed to be related to ${\tilde{u}}_{1}$. However, the dynamics in (Equation 1) reveal that the angle constraints and angular velocity constraints will also have impacts on the control input in the x axis. This may be solved by a coupled weighting matrix in future research. The results in our tests show that the proposed strategy is sufficient to improving the performance. The signal block diagram of the whole control scheme is presented in Figure 3.

## 4. Simulations

### 4.1. Implementation Details

To show the effectiveness and advantages of the proposed method, several simulations were implemented using the PX4/Gazebo platform [42] together with the Robot Operating System (ROS) middleware. The simulations are running on a PC equipped with an Intel^®^ 2.20 GHz CoreTM i7-8750H CPU (x8) and 16 GB RAM which runs the Linux Ubuntu 16.04 LTS operating system. The optimal control problem in (Equation 14) is solved by the solver qpOASES [43]. The system parameters of the FUAV platform are presented in Table 2.

### 4.2. Simulation Results

Three simulations are implemented to present the improvements and capabilities of the proposed method whose scene is shown in Figure 4. The first simulation is about a comparison of the performance during the normal push and slide task between the general impedance controller and the proposed NMPIC. In the second simulation, the ability to simultaneously handle the constraints and compliance of the proposed NMPIC is presented by adding force constraints when conducting the push and slide task. The results also show a new way for force regulation only by adding proper force constraints. In addition, the advantages of the proposed weight adaptive strategy are also illustrated by comparing with a fixed weight NMPIC method. Third, by adding more constraints, the power to ensure the safety of the proposed NMPIC is shown by implementing a challenging task during which the contact will suddenly lose.

#### 4.2.1. Normal Case

In the first case, a normal push and slide task along a smooth plane wall is applied after the steady-state hovering of the FUAV at the position $r_{d}={\left[0,0,1\right]}^{T}$. A box with length of 3 m, height of 3 m, and width of 0.2 m is employed to simulate the wall placed at position $r_{w}={\left[0.4,0,1.5\right]}^{T}$ described by the center of the box. It is assumed that all of the states and wrench variables are within the constraints in the normal case. The push and slide task is divided into three stages: the approaching–contacting stage, sliding stage, and releasing–departing stage. The stages are divided based on the desired position setpoints. In the approaching–contacting stage, the desired position setpoint gradually increases to $r_{d}={\left[1.5,0,1\right]}^{T}$ with a velocity of 0.075 m/s around the x axis in 20 s. Following this desired position setpoint, the FUAV will first approach the wall and then be contacted at some position. After that, the contact force will increase as the desired position setpoint stop increases. Then, the desired position setpoint moves to $r_{d}={\left[1.5,1,1\right]}^{T}$ with a desired velocity of 0.05 m/s along the y axis driving the FUAV to perform a sliding operation. Finally, the desired position setpoint changes to $r_{d}={\left[0,1,1\right]}^{T}$ with a velocity of 0.075 m/s along the x axis resulting in the releasing–departing stage, during which the FUAV will slowly release the contact force and leave the wall. The entire desired process is illustrated in Figure 5.

The simulations are performed between the general IC and the proposed NMPIC. For a fair comparison, the performances are conducted using the same desired position setpoint with a precise system time provided by ROS. The parameters of IC for this case are listed in Table 3. To show the improvements of the proposed NMPIC, the same parameters of the same control law are exploited to be the reference control input for the quadratic cost function in (Equation 9). Since the inequality constraints are assumed to have no effect on the control performance in the normal case, only the model constraints are added to solve the NMPIC problem. For the same reason, the weights of the cost function can be fixed in this case, given as $\Gamma =diag\left(1,1,1,10,10,10\right)$. The time step is set to 0.1 s and the prediction horizon *N* is set to 5.

The simulation results are plotted in Figure 6. Figure 6a,b show the actual position and velocity in three dimensions of both NMPIC and IC, respectively. The three stages are denoted using different colors. In the approaching–contacting stage (blue region), it can be seen that position x increases to 0.3 m in the beginning and then maintains its position due to the contact with the wall. After contacting, the external forces begin to increase as can be found in Figure 6c,d. The measured external forces are obtained by a gazebo contact plugin attached to the wall and the estimated external forces are gained by the DOs. After the forces increase to the maximum, the FUAV begins to slide along the wall resulting in the increase in the y position in the yellow region. During the sliding stage, it can be seen that the forces remain almost constant. Then, in the releasing–departing stage (green region), the external forces begin to descend until leaving the wall surface. Finally, the position x returns to 0 m. The input forces and torques of both controllers are given in Figure 6e,f. It can be seen that they have very a similar performance in this case. However, they are not exactly the same. From Figure 6c,d, it can be observed that their applied forces are indeed different during the sliding stage, in which the applied force under NMPIC is approximately 3.1 N and the one under IC is approximately 3.6 N. This may account for the modeling errors since model constraints are considered in the optimal problem of NMPIC. Furthermore, from the represented results, we can find that both IC and NMPIC work well in the normal aerial push and slide task of FUAV.

#### 4.2.2. Force Constraints

It is important to ensure safety when robots are interacting with environments, especially during sustained contact-based tasks. One of the possible strategies to guarantee safety is to limit the interaction forces. To show the capability of the proposed NMPIC under force constraints and the advantages of the proposed adaptive weight strategy, a scenario in which the FUAV is assigned to apply force to a fixed point is introduced. In addition, three controllers, including IC, NMPIC, and NMPIC with fixed weight (FNMPIC), are compared in this case.

Similarly with the normal case, the FUAV is first hovering at the position $r_{d}={\left[0,0,1\right]}^{T}$. Then, the desired position moves to $r_{d}={\left[3,0,1\right]}^{T}$ with a uniform velocity of 0.1 m/s along the x axis. The parameters of the impedance controller are all the same as those in the normal case. The fixed weight of the FNMPIC is set to $\Gamma =diag\left(10,10,10,10,10,10\right)$. For a fair comparison, the base weighting matrix $\Gamma_{b}$ in (Equation 15) is set to the same as Γ in FNMPIC. To determine the constraints-related weighting matrix, the force constraints used in this case are given as
(17)$${\left[-5,-5,16,-2,-2,-2\right]}^{T}\leq u\leq {\left[2,5,24,2,2,2\right]}^{T}$$

Since only force constraints are considered in this case, the constraints-related weighting matrix is only related to force constraints. The total number *n* in (Equation 16) of the constraints for each channel is 1. To be specific, take the force related to x axis as an example. In this case, $q_{1}=u_{1}^{*}$ and $\gamma_{c1}=w_{11}$ are the specifications of (Equation 16). Based on the constraints set in (Equation 17), the parameters for calculating $w_{11}$ are set as follows: $\Delta_{1}=0.1$, $\alpha_{1}=-5$, $\beta_{1}=2$, $\epsilon_{1}=0.05$. The parameters in five other channels are set by similar ways. The parameters $\alpha_{i}$ and $\beta_{i}$ are set based on the constraints (Equation 17). Other parameters Δ and ϵ are identically set to 0.1 and 0.05 for all channels, respectively.

The simulation results in this case are shown in Figure 7. Figure 7a presents the actual positions of the three controllers. In this task, the FUAV firstly approaches the wall and then begins to increase the force and meanwhile maintains the position. From Figure 7d, it can be seen that the interaction force increases to approximately 8 N under IC without force constraints. Thanks to the addition of the force constraints in the proposed NMPIC, the interaction force is limited at 2 N, as seen in Figure 7b. However, purely adding the force constraints will lead to an unstable problem because of the fixed weighting matrix, as plotted in Figure 7a,c. The reason for this is the conflict between the reference force and the force constraint. The reference force related to the x axis is illustrated in Figure 7f. With the help of the proposed adaptive strategy, the weight corresponding to channel of force around the x axis will decrease as the reference force enters the constraints region (blue region), as presented in Figure 7e. As a consequence, the FUAV has the power to handle the constraint and maintain stability at the same time. From Figure 7b, it can also be seen that, even though no direct force control method is used, the controller is able to regulate the force to the force constraints. In this case, the force constraints can be viewed as the desired force. This method also has the potential to perform variable force-tracking operations by online changing the force constraints.

#### 4.2.3. Loss of Contact

During the real inspection operations, some unexpected situations may occur such as an unpredictable loss of contact. This may account for the lack of prior information about the environment or the measurement faults of the localization system. To simulate this scenario, the FUAV is assigned to perform an inspection task along a wall with unknown length. For comparison, IC and the proposed NMPIC with all the position, velocity, and force constraints are considered in this case.

As illustrated in Figure 8, the FUAV is first assigned to contact point A and apply a force with 2 N along the x axis. This is obtained by moving the desired setpoint from $r_{d}={\left[0,0,1\right]}^{T}$ to $r_{d}={\left[2,0,1\right]}^{T}$, similarly to the last case. Then, the desired setpoint continues moving along the y axis with a desired velocity of 0.05 m/s. Since the length of the wall is assumed to be unknown in this case, the terminal desired setpoint is set to $r_{d}={\left[2,2,1\right]}^{T}$ to complete the inspection task of the whole wall. As a consequence, the FUAV will suffer from a loss of contact at point B during the inspection process. To prevent a collision with the wall, the state constraints are inserted together with the force constraints in the last case to solve the optimal problem (Equation 14), formulated as
(18)$$\begin{array}{c} {\left[-5,-5,-1,-1,-1,-1,-\frac{\pi }{3},-\frac{\pi }{3},-\frac{\pi }{3},-2,-2,-2\right]}^{T}\leq X\\ \leq {\left[0.45,2,2,1,1,1,\frac{\pi }{3},\frac{\pi }{3},\frac{\pi }{3},2,2,2\right]}^{T} \end{array}$$

It is worth noting that the constraints for Euler angles here are designed to avoid singularity. The position x is limited to 0.45 m to avoid bumping into the wall. This value is obtained by preliminarily acquiring the position of the wall in this case. The position can be also obtained by collision detection using the DO if it cannot be measured in advance. The parameters of IC and the reference impedance controller of NMPIC are the same as those defined in the last case. The only difference between the parameters in this case is the weighting matrix constraints-related weighting matrix $\Gamma_{c}$. Since more constraints are added, the calculation of $\Gamma_{c}$ is more complicated. Furthermore, take the force around the x axis for explanation. The adaptive strategy of (Equation 16) becomes $\gamma_{c1}=w_{11}*w_{21}*w_{31}$ with $q_{1}=x_{d}$, $q_{2}={\dot{x}}_{d}$ and $q_{3}=u_{1}^{*}$. The parameters for calculating $w_{j1},j=1,2,3$ are set as follows: $\Delta_{1}=0.1$, $\epsilon_{1}=0.05$, $\Delta_{2}=0.1$, $\epsilon_{2}=0.1$, $\Delta_{3}=0.1$, and $\epsilon_{3}=0.1$. $\alpha_{i}$ and $\beta_{i},i=1,2,3$ are set according to the corresponding values defined in (Equation 18). Similar parameters are set for the five other channels. For simplicity, the $\Delta_{i}$ and $\epsilon_{i}$ are set as the same as those in the channel of force around the x axis.

The simulation results in this case are visualized in Figure 9. In this challenging task, the FUAV first approaches the wall and is contacted, the same as that in the normal case. The position x increases in the beginning and then remains constant while contacting, as shown in Figure 9a. After that, the FUAV slides along the wall, which is also the direction of the y axis, as seen in Figure 9b. Since the length of the wall is unknown, the FUAV will suffer from a loss of contact at the edge of the wall, denoted as the blue dash line in Figure 9. From Figure 9a, it can be seen that the x position enters the red region after the loss of contact, which indicates that the body of the FUAV collides with the wall. It can also be seen in Figure 9c that the that the FUAV falls down after the collision. This could cause severe damage if it happens in a real flight. By applying the proposed NMPIC, the situation is significantly improved. As can be observed from Figure 9a, the FUAV will rush forward for a short distance because of inertia once the loss of contact occurs. After that, the position constraint of x will force the FUAV back to avoid colliding with the wall. Although some oscillations exist, the FUAV will eventually be stable at the position of 0.43 m within the constraint 0.45 m. This simulation result significantly shows the power of the proposed NMPIC for ensuring safety in the physical interaction task. In Figure 9d,e, the measured and estimated external forces are shown. After the loss of contact, it can be seen that the measured forces disappeared, but the estimated external forces still exist. This phenomenon accounts for the insensitivity of the DO to signals with high frequency. The oscillations may be attributed to this phenomenon. In Figure 10a, the changing weights in the entire process are presented. In contrast to that in the last case, the weights descend twice during the process. The first descent happens when the desired setpoint reaches the position constraint, as can be seen in Figure 10c. The second descent happens when the reference force reaches the force constraint, as can be seen in Figure 10b. These descents are implemented to balance the conflict between the reference signals and the constraints.

## 5. Conclusions

In this work, a control scheme is proposed to tackle the issue of simultaneously handling the constraints and maintaining compliant behavior in the aerial physical interaction with the environment using a FUAV. The proposed NMPIC combines the advantages of both the NMPC and general IC by treating IC as reference signals of NMPC. A DO is employed for both external wrench estimation and model compensation for the NMPIC controller. To address the conflict between the reference signals and the constraints, a weight adaptive strategy is proposed to tune the weighting matrix of the cost function online in the optimal problem of NMPIC. Three simulation scenarios are presented to validate the effectiveness and advantages of the proposed method. The results show that, in the normal case, the NMPIC can have a similar performance as a general IC. Under the force constraints, the NMPIC with the weight adaptive strategy can perform well both in restricting the interaction force and maintaining stability. By viewing the force constraints as the desired forces, the proposed method is able to regulate force without using any direct force control method. It also has potential to perform force tracking tasks by changing the force constraints online, which will be further studied in the future. Even in the case when contact is suddenly lost, the NMPIC has the capability of avoiding collision and eventually returning to stability.

## Figures and Tables

**Figure 1 sensors-23-05231-f001:**
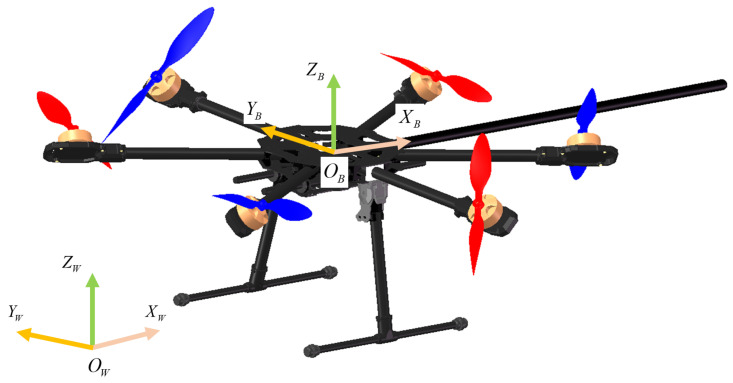
Illustration of the FUAV platform and the coordinate frames.

**Figure 2 sensors-23-05231-f002:**
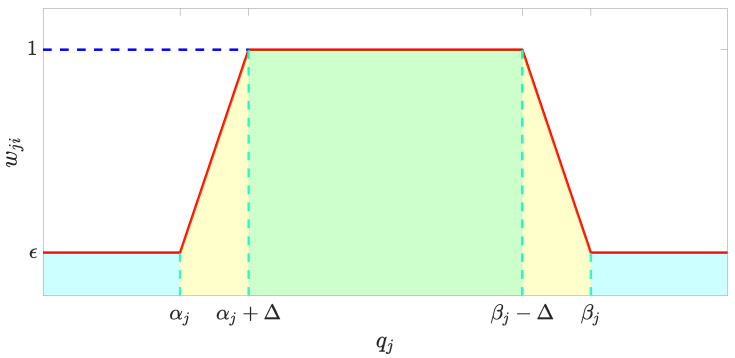
Illustration of the curve for calculating the adaptive weight. The blue region represents the values out of constraint while the green region denotes the values safely restricted in the constraint. The yellow region shows the values close to the boundary of constraints whose size is described by ϵ.

**Figure 3 sensors-23-05231-f003:**
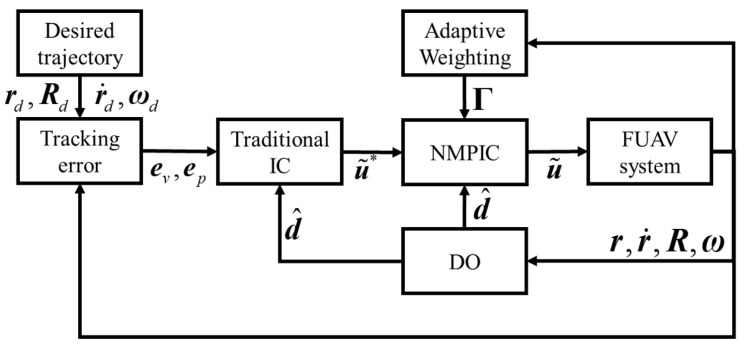
Signal block diagram of the whole control scheme.

**Figure 4 sensors-23-05231-f004:**
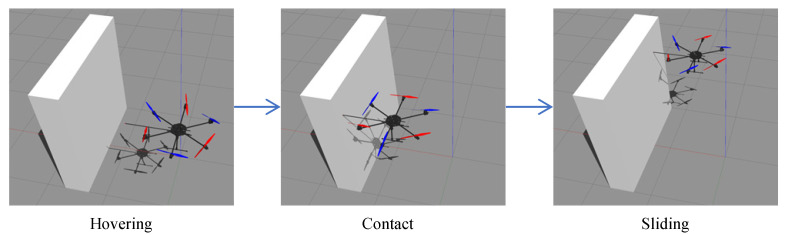
Simulation scene.

**Figure 5 sensors-23-05231-f005:**
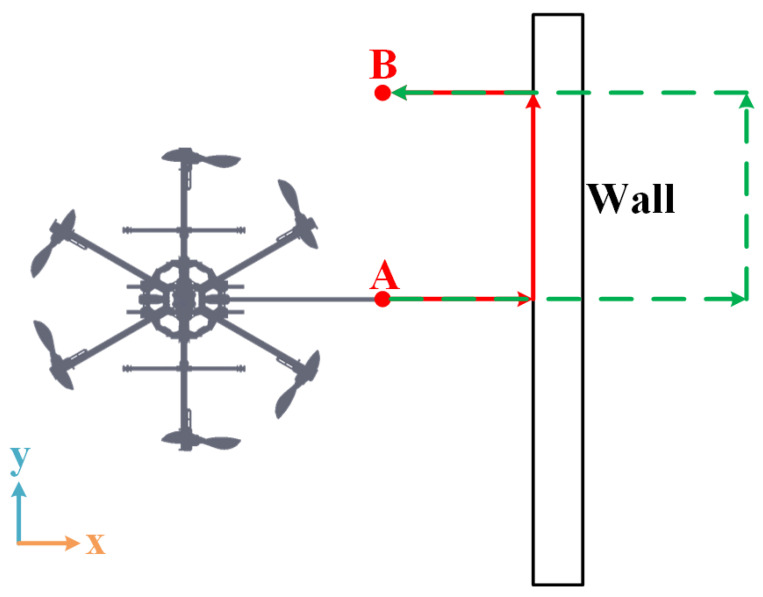
Desired trajectory for a normal case from point A to point B. The green dash line indicates the desired setpoint trajectory while the red solid line denotes the ideal flight trajectory of the tool tip.

**Figure 6 sensors-23-05231-f006:**
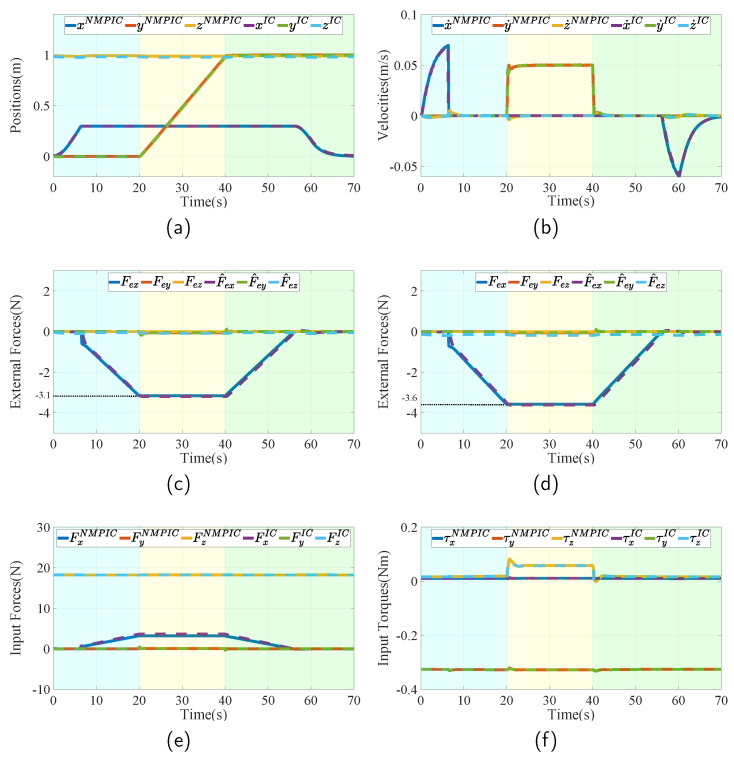
Simulation results in the normal case, during which the FUAV is assigned to conduct a normal push and slide task. The whole interaction process is divided into three stages: the approaching–contacting stage (blue region), the sliding stage (yellow region), and the releasing–departing stage (green region). The superscripts NMPIC and IC indicate which control process the parameters belong to. (**a**) Actual position of the FUAV body compared between the IC and the proposed NMPIC. (**b**) Actual velocity of the FUAV body compared between IC and NMPIC. (**c**) Measured and estimated external forces during the task controlled by NMPIC. (**d**) Measured and estimated external forces during the task controlled by IC. (**e**) Control forces compared between IC and NMPIC. (**f**) Control torques compared between IC and NMPIC.

**Figure 7 sensors-23-05231-f007:**
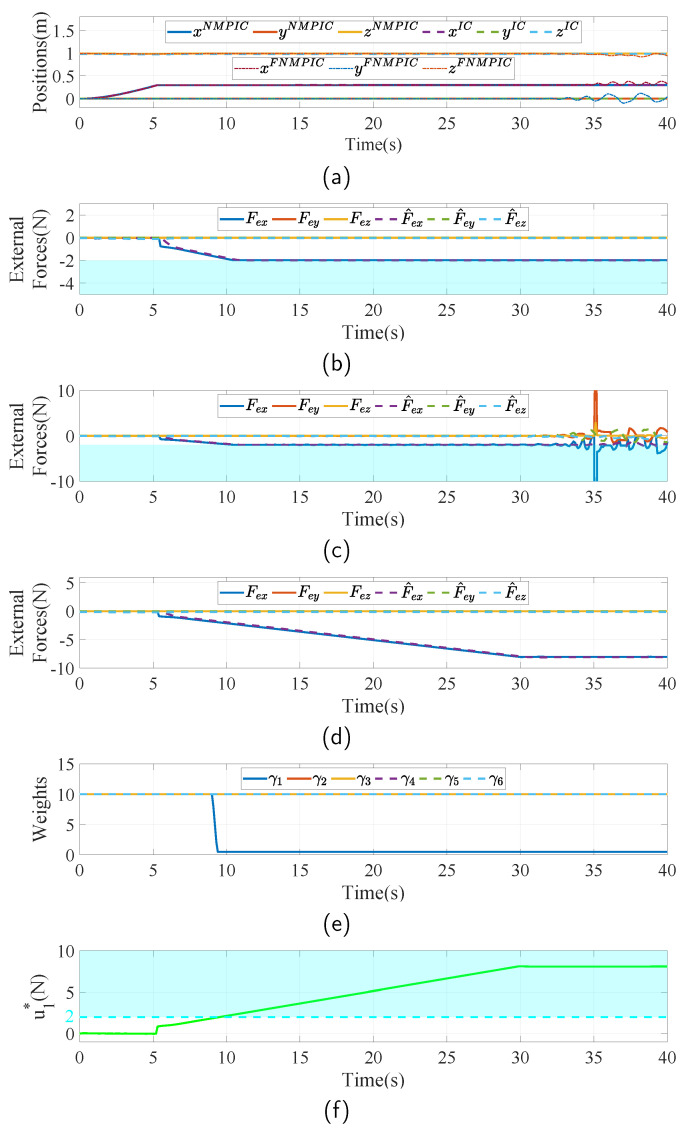
Simulation results with force constraints. The blue region indicates the range out of force constraints. (**a**) Actual position of the FUAV body among IC, NMPIC, and FNMPIC. (**b**) Measured and estimated external forces during the task controlled by NMPIC. (**c**) Measured and estimated external forces during the task controlled by FNMPIC. (**d**) Measured and estimated external forces during the task controlled by IC. (**e**) Weights–changing process based on the proposed adaptive strategy. (**f**) Illustration of the force constraint and the force reference signal around the x axis.

**Figure 8 sensors-23-05231-f008:**
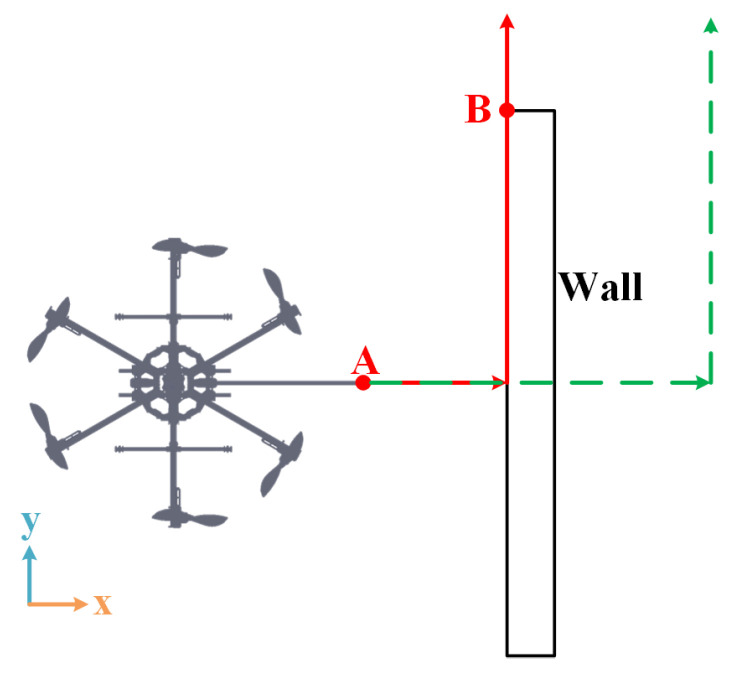
Desired trajectory for the loss of contact case starting from point A. Point B is the point where the FUAV will lose contact. The green dash line indicates the desired setpoint trajectory while the red solid line denotes the ideal flight trajectory of the tool tip.

**Figure 9 sensors-23-05231-f009:**
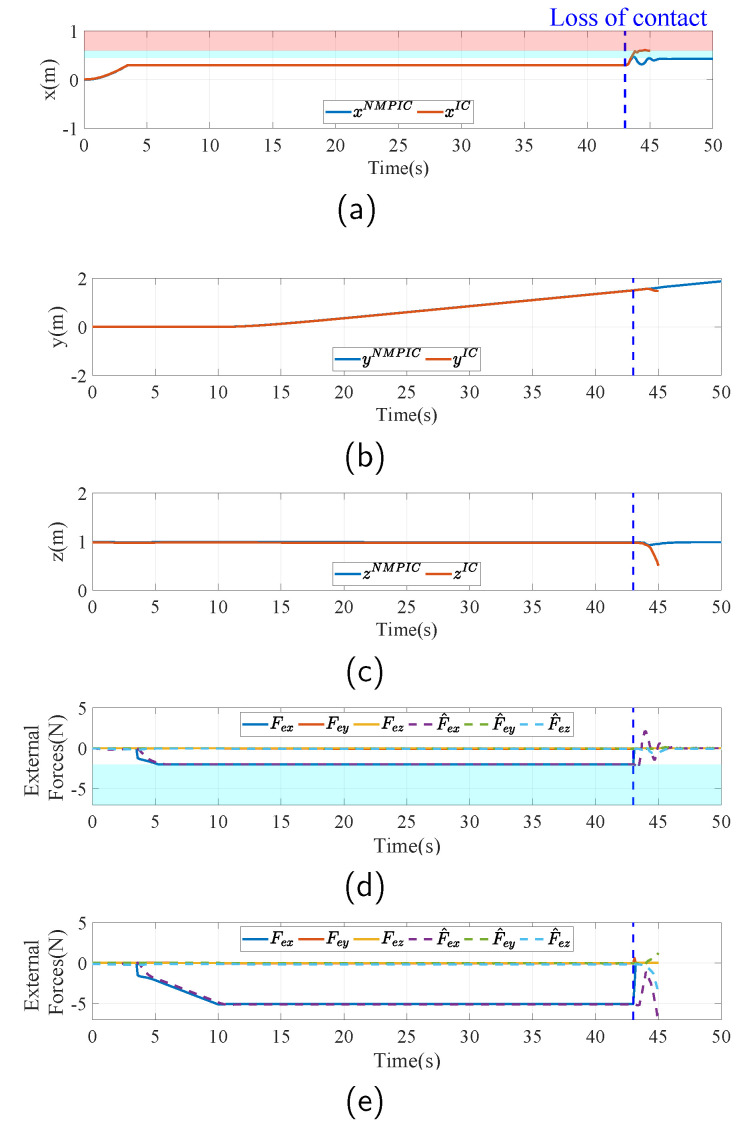
Simulation results in the case of the loss of contact. The blue dash line indicates when the loss of contact occurs. The light blue region indicates the range out of the constraints and the red region means a collision with the wall. (**a**–**c**) Actual x, y, z position of the FUAV body compared between IC and NMPIC. (**d**) Measured and estimated external forces during the task controlled by NMPIC. The blue region denotes the force constraint. (**e**) Measured and estimated external forces during the task controlled by IC.

**Figure 10 sensors-23-05231-f010:**
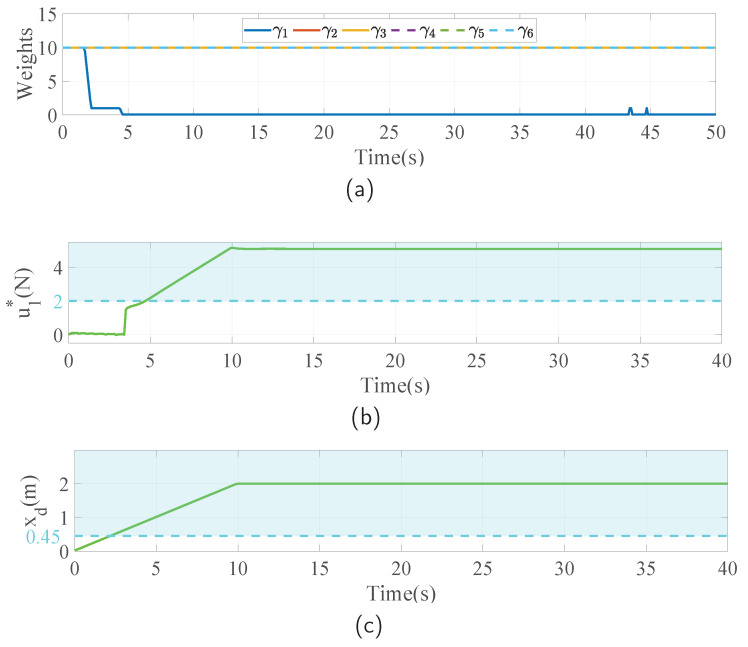
Weights changing process, reference inputs, and setpoints in the case of loss of contact. The blue dash line indicates when the loss of contact occurs. The light blue region indicates the range out of the constraints. (**a**) The weights changing process is based on the proposed adaptive strategy. (**b**) Illustration of the force constraint and the force reference signal around the x axis. (**c**) Illustration of the position constraint and the desired position signal around the x axis.

**Table 1 sensors-23-05231-t001:** The corresponding relationship between the constraints and the control channel.

Control Input	Parameter	Constraint Variable	Description
${\tilde{u}}_{1}$	$\gamma_{c1}$	$x_{d}$, ${\dot{x}}_{d}$, $u_{1}^{*}$	Force around the x axis
${\tilde{u}}_{2}$	$\gamma_{c2}$	$y_{d}$, ${\dot{y}}_{d}$, $u_{2}^{*}$	Force around the y axis
${\tilde{u}}_{3}$	$\gamma_{c3}$	$z_{d}$, ${\dot{z}}_{d}$, $u_{3}^{*}$	Force around the z axis
${\tilde{u}}_{4}$	$\gamma_{c4}$	$\phi_{d}$, $\omega_{xd}$, $u_{4}^{*}$	Torque around the x axis
${\tilde{u}}_{5}$	$\gamma_{c5}$	$\theta_{d}$, $\omega_{yd}$, $u_{5}^{*}$	Torque around the y axis
${\tilde{u}}_{6}$	$\gamma_{c6}$	$\psi_{d}$, $\omega_{zd}$, $u_{6}^{*}$	Torque around the z axis

**Table 2 sensors-23-05231-t002:** System parameters of the FUAV platform.

Parameter	Value	Unit
*m*	1.8	Kg
$J_{xx}$	0.05	$\mathrm{Kg}m^{2}$
$J_{yy}$	0.05	$\mathrm{Kg}m^{2}$
$J_{zz}$	0.09	$\mathrm{Kg}m^{2}$
*l*	0.34	m
*d*	0.04	m
α	$47^{\circ }$	deg
$c_{f}$	$2.0\times 10^{-5}$	$N/{\mathrm{Hz}}^{2}$
$c_{\tau }$	0.014	m
*L*	0.7	m

**Table 3 sensors-23-05231-t003:** Parameters of IC.

Parameter	Value
$M_{\upsilon }$	$diag\left(1.5,1.5,1.5,0.045,0.045,0.085\right)$
$D_{\upsilon }$	$diag\left(8,8,8,2,2,2\right)$
$K_{\upsilon }$	$diag\left(5,5,5,3,3,3\right)$
$L_{1}$	$diag\left(10,10,10,4,4,4\right)$
$L_{2}$	$diag\left(25,25,25,4,4,4\right)$

## Data Availability

Not applicable.

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
