# Peer review of "Nonlinear Model Predictive Impedance Control of a Fully Actuated Hexarotor for Physical Interaction"

_sensors, 2023, doi:10.3390/s23115231_

Round 1

Reviewer 1 Report

Paper presented for hexarotor performing physical interaction with the environment through a rigidly attached tool. The simulation design is interesting. A nonlinear model predictive impedance control (NMPIC) scheme is discussed to obtain the goal. A disturbance observer is exploited to estimate external wrench and then provide compensation for the model that employed in the controller. The results also indicated a good output. 

However some suggestion to improve the result.

1. It should be required to check control output with respect to disturbances and modeling errors.

2. Most reviewed papers in this work are old, however some works on Unmanned aerial vehicle/drones using fractional MIMO PID suggested most recently. These are not included.

3. Novelty with respect to PID or FOPID should be discussed as limitations before presenting the nonlinear model predictive impedance control.

4. Recent papers can be included to improve the introduction, with various strategies recent on Fractional-order MIMO quadrotors and mainly verified real-time. 

Some place flow of writing can be improved.

Author Response

Dear Reviewer,

Thanks for your detailed and helpful comments. The corresponding responses can bu found from the attachment.

Reviewer 2 Report

       The authors considered the problem of a fully-actuated hexarotor performing physical interaction with the environment through a rigidly attached tool. They have proposed the NMPIC method to achieve the goal of having the controller be able to simultaneously handle the constraints and maintain compliant behavior. The manuscript is well written, and the mathematical development is clear and objective. The following suggestions and questions are mentioned below: 

1.     The punctuation marks should be checked throughout the paper, especially after the equations and at the end of the statements.

2.     English is generally good. I think it needs to be polished further.

3.     What is meant by Impedance control?

4.     What is the purpose of using FUAV?

5.  What is meant by d(t_k) in equation (10)?

6.     Write the equation (14) in the proper format.

7.     On page 14, line 418 is incomplete.

So I suggest a Minor Revision this time

Needs improvement

Author Response

(The authors gave the same response as above.)

Round 2

Reviewer 1 Report

The paper is updated as per my previous comments. Good work.

The paper is updated as per my previous comments. Good work.